# Small Extracellular Vesicles from Hypoxic Triple-Negative Breast Cancer Cells Induce Oxygen-Dependent Cell Invasion

**DOI:** 10.3390/ijms232012646

**Published:** 2022-10-21

**Authors:** Bianca Cruz Pachane, Ana Carolina Caetano Nunes, Thais Regiani Cataldi, Kelli Cristina Micocci, Bianca Caruso Moreira, Carlos Alberto Labate, Heloisa Sobreiro Selistre-de-Araujo, Wanessa Fernanda Altei

**Affiliations:** 1Biochemistry and Molecular Biology Laboratory, Department of Physiological Sciences, Universidade Federal de São Carlos—UFSCar, São Carlos 13565-905, SP, Brazil; 2Max Feffer Plant Genetics Laboratory, Department of Genetics, University of São Paulo—ESALQ, Piracicaba 13418-900, SP, Brazil; 3Center for the Study of Social Insects, São Paulo State University “Julio de Mesquita Filho”, Rio Claro 14884-900, SP, Brazil; 4Molecular Oncology Research Center, Barretos Cancer Hospital, Barretos 14784-400, SP, Brazil; 5Radiation Oncology Department, Barretos Cancer Hospital, Barretos 14784-400, SP, Brazil

**Keywords:** hypoxia, extracellular vesicles, breast cancer, cell invasion

## Abstract

Hypoxia, a condition of low oxygenation frequently found in triple-negative breast tumors (TNBC), promotes extracellular vesicle (EV) secretion and favors cell invasion, a complex process in which cell morphology is altered, dynamic focal adhesion spots are created, and ECM is remodeled. Here, we investigated the invasive properties triggered by TNBC-derived hypoxic small EV (SEVh) in vitro in cells cultured under hypoxic (1% O_2_) and normoxic (20% O_2_) conditions, using phenotypical and proteomic approaches. SEVh characterization demonstrated increased protein abundance and diversity over normoxic SEV (SEVn), with enrichment in pro-invasive pathways. In normoxic cells, SEVh promotes invasive behavior through pro-migratory morphology, invadopodia development, ECM degradation, and matrix metalloprotease (MMP) secretion. The proteome profiling of 20% O_2_-cultured cells exposed to SEVh determined enrichment in metabolic processes and cell cycles, modulating cell health to escape apoptotic pathways. In hypoxia, SEVh was responsible for proteolytic and catabolic pathway inducement, interfering with integrin availability and gelatinase expression. Overall, our results demonstrate the importance of hypoxic signaling via SEV in tumors for the early establishment of metastasis.

## 1. Introduction

Low oxygen tissue perfusion, or hypoxia, is established in most solid tumors as a consequence of cell proliferation, increased tumor mass, and abnormal tumor vascularization [1]. Oxygen depletion, either acute, intermittent, or continuous, inhibits the ubiquitination of the hypoxia-inducible factor 1 (HIF-1), whose accumulation triggers the transcription regulation of genes involved in survival, angiogenesis, invasiveness, stemness, and resistance to chemotherapy [2,3,4,5]. These responses are often upregulated in both primary and metastatic breast adenocarcinomas, impairing overall patient survival and successful therapeutics [6,7].

Tumor cells have increased extracellular vesicle (EV) secretion, which enables intercellular communication by stably transferring biomolecules between different compartments [8,9]. In triple-negative breast cancer (TNBC), a highly aggressive breast cancer subtype that lacks receptors for estrogen (ER-), progesterone (PR-), and HER2 [8,10], normoxia-produced EVs have been highlighted in aiding pro-tumoral responses such as pre-metastatic niche establishment [1], directed migration, and organotrophic metastasis [11] and in transmitting signals for potentiating malignant properties on the tumor microenvironment [12].

In many cancer types, EV secretion is favored by hypoxic stress [8] and their protein, miRNA, and lipid cargoes can differ from normoxic EVs, which contribute to pro-tumoral behaviors in recipient cells from neighboring areas [1,13]. EVs separated from hypoxic cells increase pro-angiogenic signaling and survival in endothelial cells [14,15] and suppress T cell proliferation [16]. Upon treatment with hypoxic microvesicles, normoxic tumor cells facilitate the epithelial-to-mesenchymal transition (EMT) and metastasis [15,17], and a study found that hypoxic TNBC cells preferentially uptake hypoxic EV than normoxic EV preparations [18]. Although the impact of EV signaling in tumor cells is greatly investigated [13,18], the effects of autocrine and paracrine signaling of EVh in TNBC cells are still largely unknown.

Cell invasion involves the migration of tumor cells through adjacent tissues in combination with extracellular matrix (ECM) remodeling, dynamic focal adhesion spots, and F-actin cytoskeleton polarization. It is considered an early step of metastasis that favors intravasation, extravasation, and the colonization of a new metastatic niche [19,20]. Similarly to many complex biological processes, invasion relies heavily on cytoskeleton arrangement, involving tubulin-rich structures called invadopodia [21], and the communication between the intracellular and extracellular compartments, which may be performed by transmembrane adhesion molecules such as integrins [19]. Interestingly, invadopodia formation enhances exosome secretion, indicating a key role of these nanoscale structures in cell invasion [22].

In this study, we investigated the effects of TNBC-derived, hypoxic small extracellular vesicles (SEVh) in tumor cells in vitro, comparing the responses in the human TNBC model cell line MDA-MB-231 exposed to normoxia (20% O_2_) or hypoxia (1% O_2_). In this paper, we address (1) the phenotypic alterations elicited by SEVh; (2) the ECM remodeling by gelatinases (matrix metalloproteases MMP2 and MMP9); (3) the direct interference of SEVh in in vitro invasion; and (4) the proteomic profiling of cells after SEVh exposure. We used oxygen levels of 20% to establish the condition of normoxia, which are those routinely employed in cell culture. However, it is necessary to consider that the physiological values of oxygen found in the tissues can be much lower, usually about 5% [23]. Oxygen values below 4% induce the expression of HIF1-α, therefore characterizing a hypoxic milieu.

## 2. Results

### 2.1. SEVh Isolation and Characterization

Each EV isolation process was performed and characterized following MISEV2018 guidelines [24], and all relevant data were submitted to the EV-TRACK knowledgebase (EV-TRACK ID: EV220177) [25]. On average, EVs were isolated from 1.94 × 10^7^ ± 0.48 × 10^6^ cells in hypoxia, with 94% ± 4.3% viable cells. Under normoxic culture conditions, EVs were separated from 1.98 × 10^7^ ± 0.54 × 10^6^ cells with 96.3% ± 2.6% cell viability.

Higher concentrations of protein were detected on hypoxic EV when compared to their normoxic counterparts (Figure 1a). After normalizing the protein concentration by the final volume of each sample, we obtained a total protein content of 270.5 ± 93.9 µg in hypoxic SEV (SEVh) and 109.0 ± 41.47 µg in normoxic SEV (SEVn). Using MEV samples for comparison with a different EV population, we also found increased protein content in hypoxia (76.8 ± 42.2 µg) than in normoxia (28.8 ± 11.9 µg).

Label-free LC-MS proteomic analysis identified 10 EV biomarkers [24] (ADAM10, ALIX, CD47, CD63, CD81, CD82, CD9, FLOT-1, FLOT-2 and TSG101) in all SEV samples, with an undistinguished profile between SEVh and SEVn (Figure 1b). We validated some markers by western blotting, in which an enrichment of the proteins Alix, CD63, and flotillin-1 in SEV samples from hypoxic and normoxic conditions was observed as expected (Figure 1c). Flotillin-1 was also identified in all LEV from both groups and MEV from hypoxia. Calnexin, serving as the negative control, was only identified on cell lysate and, albeit faintly, in LEV samples that were not used in this work. None of the proteins were found on the leftover condition media for either group. Hence, our object of study, SEVh, has a simplified protein profile of Alix+/CD63+/FLOT-1+/CANX−.

Particle analysis confirmed the overall abundance of hypoxic EVs when compared to their normoxic peers (Figure 1d). After normalizing the particle count by final sample volume, hypoxia increased the total particle count of SEV by 25% (SEVh = 2.03 × 10^12^ ± 2.1 × 10^11^ particles; SEVn = 1.62 × 10^12^ ± 1.5 × 10^11^ particles). In comparison, a wider difference was observed in MEV samples (5.23 × 10^10^ ± 5.8 × 10^9^ particles in normoxia, 1.14 × 10^12^ ± 2.2 × 10^11^ particles in hypoxia). NTA analysis provided a histogram displaying MEV (Figure 1e) and SEV (Figure 1f) size dispersion by particle count, which established their nomenclature; the mode for hypoxic and normoxic MEV was 243 nm, whereas, for SEV samples, it ranged from 136–150 nm.

Transmission electron microscopy allowed for the visualization of broad range and individual MEV (Figure 1g) or SEV (Figure 1h). EV preparations presented typical morphology, adequate abundance, and sizes according to their nomenclature.

### 2.2. SEVh Proteome Is Enriched with Components of Pro-Invasive Signaling Pathways

Comparative label-free proteome of SEV samples identified a total of 2390 proteins, of which 2347 were found in hypoxia and 1278 in normoxia. A total of 1112 unique proteins were enriched in SEVh, contrasting to 43 in SEVn, and 1235 proteins were shared among both groups (Figure 2a). Principal component analysis (PCA) differentiated SEVh and SEVn, clustering replicates according to each group (Figure 2b) with principal component (PC) values corresponding to 58.7%, 22.1%, 10.1%, 5.7%, and 3.3% of the designated variance.

Univariate analysis identified 552 statistically relevant proteins between SEVh and SEVn (*p* < 0.05). The top 144 proteins (*p* < 0.01) were considered for enrichment analysis (Figure 2c; Appendix A). The most significantly upregulated protein of SEVh is ISG15 (FDR: 9.59 × 10^−6^), a ubiquitin-like protein previously linked to integrin-mediated cell invasion [26]. Other distinguished species found were mTOR (3.6 × 10^−4^) and its regulator, RPTOR (1.05 × 10^−3^), critical for controlling cancer metabolism [27], and structural proteins such as microtubule-associated proteins 1B, 4, and RB (1.15 × 10^−3^; 1.78 × 10^−3^; 7.3 × 10^−3^, respectively). Two protein families were abundant in SEVh: the SMADs (1.12 × 10^−3^), related to the activation of TGF-β pathway [28], and RGDPs (1.67 × 10^−3^), predicted to act on the nuclear pore and to aid GTPase activity [29].

Both mTOR and TGF-β signaling pathways, alongside pro-angiogenic VEGFA/VEGFR2, were significantly favored in SEVh samples. Our analysis found an enrichment of four proteins of the mTOR pathway (RRAGB, RPTOR, MTOR, and RRAGA; FDR: 2.7 × 10^−3^), five of the TGF-β pathway (SMAD2, SMAD3, SMAD9, SMAD1, and SMAD5; FDR: 3.5 × 10^−3^), and eleven of the VEGFA/VEGFR2 pathway (ABCF2, FXR2, AP2S1, SHC2, ARF6, ARF4, MTOR, CDC42BPB, STAM, SHC1, and EIF3H; FDR: 3.7 × 10^−3^). We identified that pathways related to vesicle-mediated transport (6.7E-03), RNA metabolism (1.5 × 10^−2^), DNA replication (3.01 × 10^−2^), and cell cycle (0.21) were statistically upregulated in SEVh (Figure 2e).

### 2.3. SEV-h Signaling Triggers Long-Term MDA-MB-231 Invasion

SEVh interference with TNBC invasion was assessed in vitro using two methods: (i) a transwell invasion assay with Matrigel coating (Figure 3a), and (ii) a fluorescent gelatin-covered coverslip, in which invasion was assessed by matrix degradation (Figure 3b). Matrigel invasion was induced by hypoxia alone and by SEVh treatment in normoxia after 16 h, with non-statistical tendencies observed as early as 6 h (Figure 3a and Appendix A). Interestingly, invasion was not induced in hypoxic cells exposed to SEVh, suggesting that oxygen may either be necessary to promote SEVh-mediated invasion, or its excess may induce redox imbalance. Similar results were observed on gelatin invasion, where matrix degradation was not promoted by SEVh treatment in hypoxia (Figure 3b), thus highlighting the importance of oxygen for SEVh-mediated invasiveness.

SEVh signaling in normoxia was a driving force for gelatin invasion after 48 h, as was hypoxia (FBS- group), with non-statistical tendencies appearing after 24 h (Figure 3b and Appendix A). The size and abundance of degradation spots on the fluorescent matrix varied between groups (Figure 3c) and was more pronounced in normoxic SEVh-treated cells (Figure 3d). We expected spots to be presented on the cell periphery, yet, despite the abundance of degradation in this group, few cells colocalized with them, which suggests that cellular motility was favored in this setting.

Since gelatinase activity was detected, we investigated the effects of SEVh and/or hypoxia in MMP-2 and MMP-9. Our results suggest a significant increase in the activity of soluble MMP-2 (Figure 3e) and MMP-9 (Figure 3f) 48 h after SEVh exposure in normoxia. Intriguingly, MMP-2 was detected as a 52 kDa band on SEVh-treated groups, whereas a regular 62 kDa band was found on positive controls (FBS+). No gelatinase activity was detected after 24 h (Appendix A). Regarding gene expression, SEVh signaling and/or hypoxia drastically increased MMP-2 (Figure 3g), whereas MMP-9 was diminished only after SEVh exposure in hypoxia (Figure 3h). Gene expression results were not corroborated by protein expression; MMP-2 (Figure 3i) was boosted only in hypoxic SEVh-treated cells, while MMP-9 protein expression (Figure 3j) did not vary significantly among groups. These results suggest that SEVh and hypoxia signaling are interfering in separate regulatory mechanisms of gelatinase expression.

### 2.4. SEVh Exposure Favors Migratory Morphology of MDA-MB-231 Cells in Normoxia

Upon visual interpretation, cell morphology was modified slightly by SEVh treatment in all conditions, particularly on the Matrigel coating (Figure 4a). Untreated cells in normoxia have a distinct F-actin arrangement, with parallel fibers crossing the cytoplasm on its axis. This pattern is disrupted by hypoxia and/or SEVh, which generates a specific F-actin arrangement alongside the cellular perimeter. The same behavior is visible in uncoated and gelatin-coated groups. An interesting feature of SEVh signaling in normoxia is the development of invadopodia, F-actin-rich protuberances displayed throughout the periphery of cells [30] (Figure 4b).

Overall, cell circularity decreased after exposure to collagen-based matrices for 24 and 48 h (Figure 4c–e). In uncoated surfaces, SEVh treatment in hypoxia and normoxia increased the circularity of cells compared to their respective controls in both timepoints (Figure 4c). When collagen-based matrices were used, this behavior changed; after 24 h, SEVh-treated normoxic cells were 32% more elongated in gelatin than the untreated control (Figure 4d), and they were 58% more elongated in Matrigel (Figure 4e). For comparison, cell circularity was not altered by hypoxia alone, and it was only changed after 24 h exposure to SEVh in hypoxia when cells were adhered to Matrigel (Figure 4e). Considering that increased circularities may result from cell stress, our results suggest that any type of collagen-based ECM is essential for the development of an elongated phenotype in TNBC.

Since there is an inherent phenotypical diversity of MDA-MB-231 cells, which morphs according to matrix stiffness and function [31], cell circularity should not be the only parameter to compare cell phenotypes. Hence, to confirm migratory morphology, we cross-referenced circularity results with auxiliary parameters of AR and perimeter/area (Appendix A–f). Matrigel-adhered SEVh-treated normoxic cells had higher AR and perimeter/area values, indicating lengthier cells and confirming a migratory phenotype.

### 2.5. Hypoxia Interferes with β1 and β3 Integrin Subunit Expression in TNBC Cells

SEVh treatment did not alter the gene and protein expression of integrin subunits β1 and β3, which regulate migration and invasion [32]. On the other hand, hypoxia appeared to be a keen regulator for integrin subunit β1. Hypoxia significantly upregulated gene expression (Figure 4f) and protein expression in whole cell lysates (Figure 4g). However, when probing for peripheric integrin levels via flow cytometry, we found that surface-available β1 was diminished in hypoxia (Figure 4h).

The same promoting effect of hypoxia towards the integrin gene and protein expression was found for the β3 subunit; however, there was no downregulation of surface-available β3 integrin (Figure 4i–k). Although SEVh signaling did not interfere with integrin β3 expression in normoxia, it decreased gene expression by 36.3% in hypoxia (Figure 4i). Interestingly, it did not alter protein levels on whole cell lysates (Figure 4j) but dwindled surface-available β3 to basal levels (Figure 4k).

### 2.6. Hypoxia-Derived Apoptosis Is Modulated by SEVh Signaling

The management of cell death by either apoptosis, necrosis, autophagy, or other processes in hypoxia is an established response to hypoxia [33], which we corroborated. The exposure of cells to 1% O_2_ atmosphere increased the late apoptosis percentage when compared to the untreated normoxic control (Figure 5a,b). Other parameters analyzed (early apoptosis, live and dead cells) were unchanged between groups. Focusing on SEVh signaling, the lack of statistically relevant differences in cell fate under both oxygen settings indicates that SEVh (1) aids the modulation of hypoxia-derived apoptosis response and (2) does not promote cell death in normoxic cells.

Hypoxic cells regulate apoptosis via the HIF pathway, whose main component (HIF-1) controls the balance between pro- and anti-apoptotic signals [34]. HIF-1 is a heterodimer formed when an α subunit evades ubiquitin degradation and assembles with a β subunit [34]. Our findings indicate that HIF-1α gene expression is diminished in MDA-MB-231 cells in hypoxia, regardless of SEVh exposure (Figure 5c). Although not statistically relevant, we also saw a tendency for decreased HIF-1α gene expression after SEVh signaling, especially in hypoxia (*p* = 0.07). Its protein levels followed the same pattern from the gene expression analysis, but with a significant increase in HIF-1α in SEVh-treated hypoxic cells (Figure 5d). The paradoxical result in this group may suggest that SEVh is also interfering with the regulatory mechanism of HIF-1α when oxidative stress is established.

### 2.7. MDA-MB-231 Comparative Proteome Shows That SEVh Favors a Shift in Cytoskeleton Arrangements and Cell Cycle in Normoxia

The label-free proteomic analysis of cells exposed to SEVh, under normoxia and hypoxia conditions, showed a distinct proteome profile for each group. Overall, a total of 2615 proteins were detected, with 1441 shared among all groups. We found 11 proteins exclusive to normoxic groups and 124 in hypoxia, with 196 unique proteins in untreated cells in normoxia (PBS-N), 18 in SEVh-treated cells in normoxia (SEVh-N), 108 in untreated cells in hypoxia (PBS-H), and 120 in SEVh-treated cells in hypoxia (SEVh-H). Shared proteins between groups are described in the Venn diagram in Figure 6a. Principal component analysis (PCA) clustered replicates according to their determinant characteristics (Figure 6b). PC values correspond to 36.0%, 20.7%, 11.8%, 10.2%, and 4.8% of the designated variance.

Univariate statistical analysis identified 105 proteins with differential expression between all four groups (*p* < 0.05), with the 62 most divergent proteins (*p* < 0.01, Figure 6c, Appendix A). Pairwise comparisons detected 79 distinct proteins between untreated cells in normoxia versus hypoxia (PBS-N vs. PBS-H), 66 for SEVh-treated cells in normoxia versus hypoxia (SEVh-N vs. SEVh-H), 18 for untreated versus SEVh-treated cells in normoxia (PBS-N vs. SEVh-N), 42 for untreated versus SEVh-treated cells in hypoxia (PBS-H vs. SEVh-H), and 80 for untreated, normoxic cells versus SEVh-treated, hypoxia cells (PBS-N vs. SEVh-H). The top 62 divergent protein abundance values are displayed in Figure 6d, and the interaction diagram of all statistically distinct proteins is in Figure 6e.

SEVh directly interferes with the proteome of cells in normoxia, especially with the upregulation of γ-tubulin complex binding by TUBGCP6, TUBG1, TUBGCP3, and TUBGCP2 (FDR: 9.7 × 10^−6^), hence the increasing microtubule nucleation (FDR: 2.6 × 10^−4^) and protein polymerization (FDR: 3.3 × 10^−4^). Other additional pathways induced by SEVh signaling include mitochondrial biogenesis, the citric acid cycle, and the synthesis of ubiquitin E1 and E2 enzymes (FDR: 1.71 × 10^−1^). All together, these results indicate that SEVh signaling in normoxia promotes substantial alterations in cell metabolism, favoring the cell cycle and survival of tumoral cells.

In hypoxia, SEVh treatment led to distinct responses, favoring response to stress (KEAP1, RBX1, POLR2C, POLR2I, ADD1, SETD7, NPLOC4, NCCRP1, UCHL5, CTH, BAG6, HSPA2, AK4, C4orf27, and POLR2E; FDR: 1.82 × 10^−2^) and other metabolic processes (FDR: 2.39 × 10^−2^). Particularly, it led to an increase in mRNA biogenesis and splicing (PQBP1, POLR2C, POLR2I, DDX23 and POLR2E), DNA repair (RBX1, POLR2C, POLR2I, NPLOC4 and POLR2E), and the degradation of cysteine and homocysteine (FDR: 8.04 × 10^−2^). Overall, these results reveal an oxygen-dependent modulation for SEVh signaling in tumoral cells, generating distinct survival mechanisms.

Hypoxia alone also interferes with the cell metabolism, increasing particularly the catabolic activity of both SEVh-treated (6.9 × 10^−3^) and untreated cells (6.02 × 10^−3^) and the response to stress of the control groups (3.34 × 10^−1^). Although programmed cell death, a staple of hypoxia signaling, was slightly favored by our proteomic analysis, our results also show an increase in autophagy (FDR: 3.34 × 10^−1^ for untreated groups; 2.5 × 10^−1^ for SEVh-treated groups). Akin to the literature, several proteins from hypoxic cells are related to DNA damage and repair, including USP7, BABAM1, RBX1, and POLR2I (3.34 × 10^−1^). On the other hand, SEVh signaling induces major mitochondrial and proteolytic pathways in hypoxia when compared to normoxia, such as the transport and synthesis of PAPS (8.99 × 10^−2^) and metalloprotease deubiquitinating enzymes (DUBs) (2.05 × 10^−1^). Details of enriched biological pathways may be found in Figure 6f.

## 3. Discussion

Cell invasion is critical for triggering the metastatic cascade since it allows cells to detach from the primary tumor and seek a new niche for development [35]. Intratumoral hypoxia is one of the promoting factors of this behavior, whose establishment is a direct consequence of increased proliferative rates and nutrient consumption [36]. While hypoxia alone is crucial for eliciting pro-tumoral responses, here we demonstrated that its derived SEV may also contribute to tumor progression and invasive response in TNBC cells, particularly under oxygen-rich settings.

Our study focuses on EV isolated from cell cultures, which were characterized to discriminate between EV subpopulations as they vary in size, composition, density, and other overall characteristics [9,24]. Pure EV preparations are difficult to obtain, as current isolation strategies are flawed, hence our need to characterize them, especially when optimizing a well-established method such as ultracentrifugation [9]. Although our EV separation was suboptimal due to the possibility of the co-isolation of protein aggregates, lipid rafts, and other non-EV particles [24], our thorough characterization indicates an enrichment in SEV in our last precipitate, with minimum contamination.

To investigate the influence of SEVh in TNBC cells, we established an experimental design that cross-examined the effects expected in a tumoral setting, where tumor cells from a hypoxic area communicate through EV with its surroundings, including both normoxic and other hypoxic cells [13]. Although long-distance communication is a staple function of EV, it is not exclusive, and cases of proximal signaling have been reported [37]. Furthermore, it is important to denote that our experimental design introduced TNBC cells to a 1% oxygen atmosphere right after plating, which is distinct from most literature findings, who either vary the oxygen ratio from 0.1 up to 5% [3,38,39,40,41], submit cells to chemical hypoxia [42], or incubate adhered cells to a hypoxic atmosphere [4,17,36]. Hence, some responses we observed in this study may differ slightly from the literature, such as integrin [43] and HIF-1α [44] profiling, while others—such as increased SEV yield [8], apoptosis [45] and cell invasion [17]—are persistent. Nevertheless, this is the first study that explains how the self-signaling of SEVh to hypoxic TNBC cells lead to important intracellular modifications, including increased proteolysis, catabolism, and, overall, a severe response to hypoxic stress.

The distinct pattern of responses elicited in SEVh-treated TNBC cells in hypoxia and normoxia denotes the importance of oxygen-dependent responses in cancer, affecting specifically the HIF-1 pathway. In healthy tissues, HIF-1 is hydroxylated at a conserved proline residue, which is a mark for proteasomal degradation; however, in hypoxia, reactive oxygen species (ROS) inhibit prolyl-hydroxylase domain enzymes (PHDs) from cleaving this residue, thus enabling the binding of HIF subunits to form a transcriptional regulator that interferes with metabolism, redox homeostasis, angiogenesis, tumorigenesis, and inflammation [46]. Our results identified a downregulation of both the gene and protein expression of HIF-1α in TNBC cells exposed to hypoxia, which may have occurred due to the disruption of its two transactivation domains (TAD) located at the N- and C- terminals. While N-TAD stabilizes HIF-1α against proteasomal degradation, the C-TAD regulates gene expression alongside co-factors [47]. Interestingly, SEVh-treated hypoxic cells displayed reduced HIF-1α gene expression but upregulated protein levels, suggesting a regulatory role of SEVh in this protein expression. This way, as seen in our proteomic analysis, diminished oxygen levels regulate cancer metabolism, favoring catabolism and response to stress over other cell behaviors.

Interestingly, the protein expression of HIF-1α was also decreased by SEVh treatment in normoxia, indicating the importance of oxygen in regulating HIF-mediated cell responses. Although HIF-1 activation promotes pro-tumoral responses such as ECM remodeling—either by inducing ECM protein secretion [48] or controlling MMP expression [49]—its downregulation at a protein level led to the modulation of gelatinases. Regarding MMP-9, its protein expression remained unchanged; there was a slight downregulation in gene expression in hypoxic treated cells, yet its activity increased significantly once cells were exposed to SEVh. On the other hand, MMP-2 appeared most affected by hypoxia and/or SEVh exposure: while gene expression was upregulated in all treatments, protein expression was only higher in both hypoxic groups, with potential activity favored in SEVh-treated groups. This balanced response may occur due to their shared, yet distinct, activation pathway via MMP-1. MMP-1 is a collagenase with direct connections to NF-κB signaling that participates in the canonical activation of MMP-9 [50] and in one of the less effective, non-canonical activation pathways of MMP-2 [50]. Considering that MMP-9 can be inhibited by TIMP-1, which is favored by HIF signaling and has been previously linked to triggering pro-metastatic behavior alongside miR-210, we proposed that MMP-2 activation may be preferred over MMP-9 whenever hypoxia is present—either via their EVs or by direct oxygen stress [51].

ECM remodeling interferes directly with the expression pattern of integrins, cellular adhesion receptors that mediate outside-in and inside-out responses [32]. Integrin profiling in cancer cells differs based on tumor aggressiveness, cellular origin, and tumoral progression [52], and this profile can be passed onto extracellular vesicles [53], thus aiding vesicular trafficking [54], organotropism [11], and metastasis [55]. β1 and β3 integrin subunits are often upregulated in cancer and linked to responses of cell motility [56], adhesion [57], invasion [58,59], and angiogenesis [60,61]; hence, we investigated whether they are affected by SEVh exposure. We found that a hypoxic atmosphere, instead of the signaling from SEVh, modulates the gene and protein expression of β1 and β3 integrins; however, as seen from our flow cytometry analysis, the availability of these integrins to the cell surface in hypoxia differs in the presence of SEVh. We suggest that SEVh signaling regulates integrin trafficking pathways, preventing certain subunits, such as integrin β3, from reaching the cell surface in low oxygen settings.

A consequence of integrin differential expression and availability is an impact in cellular morphology due to the link between the cytoplasmic tail of the β subunit to F-actin via talin-1. Therefore, when the extracellular segment binds to ECM components, molecular clutches are formed to propel the cell onwards, thus creating a leading edge and promoting a migratory phenotype [20,32,62]. TNBC cells display a plethora of phenotypes based on ECM composition and stiffness [63], as observed on our morphological analysis, where the three complementary parameters identified several possibilities for cell phenotype. In a complex ECM setting (i.e., Matrigel), we observed that SEVh exposure under normoxia led to low circularity and higher elongation in TNBC cells, which is consistent with a migratory phenotype. This, coupled with the identification of invadopodia—F-actin-rich protrusion on the cellular edge [30]—and the results obtained from both in vitro invasion assays, indicates that an invasive phenotype is favored in TNBC cells by SEVh signaling whenever advantageous conditions of oxygen and matrix composition are found.

We unveiled the baseline pathways and proteins favored by SEVh signaling and/or hypoxia in MDA-MB-231 cells plated in uncoated settings, which may differ from investigations in which cells are adhered to ECM components. We found that (1) each group presented a distinct protein and pathway enrichment and (2) the proteome of SEVh-treated groups diverged from the raw proteome of SEVh, which indicates a likely internalization of these particles to elicit cellular responses. In particular, the activation of survival pathways, the modulation of the cell cycle by SEVh in normoxia, and the increase in proteolysis and DNA damage in hypoxia confirm how oxygenation is important for determining cell fate whenever SEVh signaling is available. While the proteomes of both TNBC cell line MDA-MB-231 [64] and its derived EVs [12] are well known, and an in-depth investigation towards the effects of mild (1.2% O_2_) and harsh (0.2% O_2_) hypoxia was recently published [41], here we demonstrated for the first time how SEV derived from hypoxic TNBC cells modify the proteome of their originating cells in two oxygen settings (normoxia and hypoxia), while also characterizing the proteome of said SEVh and distinguishing it from their normoxic counterparts.

Nevertheless, our study has limitations that should be addressed. We based our study on a single TNBC cell line; therefore, further investigation in different systems is necessary to unveil a thorough mechanism for SEVh-mediated invasion. The investigation focused on SEVh isolated by a differential ultracentrifugation method that, although characterized at length, has disregarded other EV subtypes and preparations with additional methods for increased EV purity. We also did not investigate oxidative stress in our cellular experiments, nor EV tracking to determine how they interact with cells. Lastly, we did not cultivate cells at a physiological oxygen atmosphere, therefore rendering our normoxic groups closer to hyperoxia and hindering our ability to translate basic research to clinical practice. Since oxygen levels in normal tissues can be as low as 5%, further studies are necessary to determine if the same results would be obtained using tumor cells cultured in this condition. Furthermore, due to the low diffusion rate of oxygen in culture medium, the equilibrium oxygen concentration in culture medium is usually much lower than in the incubator atmosphere [65]. Therefore, our results must be interpreted carefully, taking these factors in consideration.

## 4. Materials and Methods

### 4.1. Cell Lines and Culture Conditions

Triple-negative breast cancer cells (MDA-MB-231) were purchased from ATCC and maintained in DMEM supplemented with glucose (4.6 g/L), sodium pyruvate (10 mg/L), and fetal bovine serum (10% FBS) at 37 °C, 5% CO_2_. A hypoxic atmosphere (1% O_2_, 5% CO_2_, 37 °C) was set in an H35 Hypoxystation (Don Whitley Scientific, Bingley, UK), and cells were incubated immediately after plating and for no longer than 4 days. EV-depleted FBS (UC-FBS) was collected after ultracentrifugation (18 h, 100,000× *g*, 4 °C—Type 45Ti rotor, Optima XE-90 ultracentrifuge, Beckman Coulter, Pasadena, CA, USA) and filtration through a 0.22 µm syringe filter.

### 4.2. EV Isolation by Differential Ultracentrifugation

MDA-MB-231 (1.6 × 10^4^ cells/cm^2^) were plated in DMEM 10% UC-FBS and incubated in hypoxia for 24 h. Culture media were exchanged to Opti-MEM reduced serum media (Gibco) for a further 48-h incubation. Conditioned media were collected and submitted to a 4-step differential centrifugation process in order to sediment live cells (300× *g*, 10 min, 4 °C; F-35-6-30 rotor, Eppendorf 5430R), large-EV (LEV—2000× *g*, 30 min, 4 °C; F-35-6-30 rotor, Eppendorf 5430R), medium-EV (MEV—10,000× *g*, 30 min, 4 °C; Type 45Ti rotor, Beckman Coulter Optima XE-90), and small-EV (150,000× *g*, 2 h, 4 °C; Type 45Ti rotor, Beckman Coulter Optima XE-90). EV pellets were collected, washed, and resuspended in filtered PBS. This procedure was replicated under normal culture conditions (~20% O_2_, 5% CO_2_, 37 °C) for comparison.

### 4.3. Nanoparticle Tracking Analysis

EV size and abundance were assessed on Nanosight NS300 (Malvern Panalytical, Malvern, UK) with NTA software (version 2.3, build 0033). EV samples were diluted in ultrapure water (1:1000 to 1:4000) and analyzed by five 60-s captures, with the following parameters: screen gain: 2.0, camera level: 14, blur: auto, max jump distance: 14.6, min track length: auto. For processing, we used a screen gain of 10.0 and a detection threshold of 4.0.

### 4.4. Transmission Electron Microscopy (TEM)

EVs were diluted (1:2) in PBS and deposited in copper grids covered with formvar and carbon (Lot 051115, 01800 F/C, 200 mesh Cu, Ted Pella Inc.) for 20 min, RT. Samples were fixed in 2% paraformaldehyde (PFA) in 0.2 M PBS, pH 7.4 for 20 min, RT, followed by extensive washes in deionized water. Grids were exposed to 4% Uranyl-acetate (pH 4) and 2% methylcellulose solution for 10 min on ice and in the dark [66]. After drying, samples were visualized in an FEI TECNAI G^2^ F20 HRTEM microscope on 40,000× magnification.

### 4.5. Protein Quantification

Isolated EV samples, cell lysates, and conditioned media were quantified using either a microBCA kit (Thermo Fischer Scientific, Waltham, MA, USA) or a standard BCA assay kit (Thermo Fisher Scientific), according to the manufacturer’s instructions. Assays were considered successful if the BSA standard curve reached R^2^ <0.99.

### 4.6. Experimental Design

In a 6-well plate, MDA-MB-231 cells (10^6^ cells/well) were plated in DMEM 10% UC-FBS and incubated for adhesion in hypoxia (1% O_2_) or normoxia (~20% O_2_) for 24 h. Cell media were replaced under sterile conditions to contain SEVh (5 µg/mL), and cells were once again incubated in hypoxia or normoxia for 24 h prior to collection. Untreated controls (i.e., PBS) were kept in parallel.

### 4.7. Cell Lysate

Cells were collected by scraping in ice-cold PBS, pelleted (7800 rpm, 12 min, 4 °C), and resuspended in lysis buffer (50 mM Tris-HCl, pH 7.4; 1% tween-20, 0.25% deoxycholate, 150 mM NaCl, 1 mM EDTA, 1 mM sodium orthovanadate, 1 mM sodium fluoride, 0.1 mM PMSF, 1 µg/mL aproptinin, and 1 µg/mL leupeptinin) for 2 h on ice. Samples were centrifuged (14,000 rpm, 20 min, 4 °C) to collect the supernatant.

### 4.8. Western Blotting

Samples (10 µg) were mixed with Laemmli sample buffer, boiled (5 min, 100 °C), and applied to SDS-PAGE gels with Precision Plus Dual Color (Bio-Rad, Hercules, CA, USA) as the loading control. Protein samples were separated by electrophoresis, transferred to nitrocellulose membranes (0.45 µm, Bio-Rad), and stained with Ponceau S dye for 2 min for quality control. Non-specific bindings were blocked with 3% BSA in Tween-TBS buffer (140 mM NaCl, 2.6 mM KCl, 25 mM Tris pH 7.4, 0.05% Tween 20) for 1 h prior to probing with primary antibodies for: Alix (1:1000, 186,429 Abcam, Cambridge, UK), Calnexin (1:1000, mAb 2679 Cell Signaling, Danvers, MA, USA), CD63 (1:1000, 59,479 Abcam), Flotillin-1 (1:1000, 61,020 BD Biosciences, Haryana, India), HIF-1α (1:1000, 51,608 Abcam), integrin β1 (1:2000, 179,471 Abcam), integrin β3 (1:1000, 119,992 Abcam; 1:1000, 34,409 Abcam), MMP-2 (1:1000, 92,536 Abcam), and MMP-9 (1:500, 38,898 Abcam). Appropriate secondary antibodies were applied for 1 h: IgG Goat anti-Mouse HRP (1:10,000, 97,040 Abcam) and IgG Goat anti-Rabbit HRP (1:10,000, 97,051 Abcam; 1:15,000, 205,718 Abcam). Membranes were revealed with ECL substrate (1,705,061 Bio-Rad; 34,096 Thermo Fisher Scientific) for 3 min, scanned on a ChemiDoc imaging system (Bio-Rad), and analyzed using FIJI [67]. For constitutive protein probing, membranes were either cut or stripped with glycine, 0.1% SDS and 1.0% Tween 20 buffer (pH 2.2), washed, blocked, and exposed to anti-GAPDH (1:10,000, 181,602 Abcam).

### 4.9. Protein Extraction

Approximately 200 µg of protein was applied to the Amicon^®^Ultra 0.5 column (3000 NMWL, Millipore), centrifuged (14,000× *g*, 30 min, 4 °C), concentrated (1000× *g*, 2 min, 4 °C), and re-quantified (BCA). Proteins were solubilized with RapiGest SF (0.2%, Waters) at 80 °C for 15 min, followed by the addition of dithiothreitol (100 mM, Bio-Rad) for 30 min, 60 °C and iodoacetamine (300 nM, GE) for 30 min, RT. After overnight digestion in trypsin solution in 50 mM NH_4_HCO_3_ (0.05 µg/µL) at 37 °C, samples were transferred to MS vials and lyophilized.

### 4.10. Label-Free LC-MS/MS Proteomics

LC–MS was performed on a NanoElute system (Bruker Daltonik, Bremen, Germany) coupled online to a hybrid TIMS-quadrupole TOF mass spectrometer [68] (Bruker Daltonik timsTOF Pro) via a nano-electrospray ion source (Bruker Daltonik Captive Spray). For long gradient runs (2 h total run), approximately 200 ng of peptides was separated on an Aurora column 25 cm × 75 µm ID, 1.9 um reversed-phase column (Ion Opticks, Fitzroy, Australia) at a flow rate of 300 nL min^−1^ in an oven compartment heated to 50 °C. To analyze samples from whole-proteome digests, we used a gradient starting with a linear increase from 2% B to 17% B over 60 min, followed by further linear increases to 25% B in 30 min and to 37% B in 10 min, and finally to 95% B in 10 min, which was held constant for 10 min. The column was equilibrated using 4 volumes of solvent A. The mass spectrometer was operated in data-dependent PASEF [69] mode with 1 survey TIMS-MS and 10 PASEF MS/MS scans per acquisition cycle. We analyzed an ion mobility range from 1/K_0_ = 1.6 to 0.6 vs. cm^−2^ using equal ion accumulation and ramp time in the dual TIMS analyzer of 100 ms each. Suitable precursor ions for MS/MS analysis were isolated in a window of 2 Th for *m*/*z* < 700 and 3 Th for *m*/*z* > 700 by rapidly switching the quadrupole position in sync with the elution of precursors from the TIMS device. The collision energy was lowered stepwise as a function of increasing ion mobility, starting at 20 eV for 1/K_0_ = 0.6 vs. cm^−2^ and 59 eV for 1/K_0_ = 1.6 vs. cm^−2^. We made use of the *m*/*z* and ion mobility information to exclude singly charged precursor ions with a polygon filter mask and further used “dynamic exclusion” to avoid the re-sequencing of precursors that reached a “target value” of 20,000 a.u. Ion mobility dimension was calibrated linearly using three ions from the Agilent ESI LC/MS tuning mix (*m*/*z*, 1/K_0_: 622.0289, 0.9848 vs. cm^−2^; 922.0097, 1.1895 vs. cm^−2^; and 1221.9906, 1.3820 vs. cm^−2^).

### 4.11. Bioinformatic Analysis and Data Enrichment

MS scans were cross-referenced with UniProt Annotated Complete Human Proteome [70] using MaxQuant [71]. Data processing in Perseus [72] removed reverse proteins, possible contaminants, and single-sample proteins. Spectra values were normalized by total ion count (TIF = 10^6^ × intensity/∑ total intensity), transformed by −log10, and scaled following Paretto’s parameter. Principal component analysis (PCA), hierarchical clustering, and univariate statistical analysis were performed in MetaboAnalyst (v. 5.0) [73]. Data enrichment and functional analysis were executed on the databases STRING [74] and Reactome [75] (minimum requirements: medium confidence interaction score = 0.4; *p* < 0.05). False discovery rates (FDR) were considered as confidence requirements for enrichment analysis. Mass spectrometry proteomics data were deposited to the ProteomeXchange Consortium (http://proteomecentral.proteomexchange.org, accessed on 10 March 2020) via the PRIDE partner repository [76] with the dataset identifier PXD035244.

### 4.12. Gelatin Zymography

Samples (10 µg) were prepared with 50% non-reducing sample buffer and loaded into 10% gelatin (100 µg/mL) SDS-PAGE gels for electrophoresis at 4 °C. Gels were washed with 2.5% Triton X-100 for 40 min, RT, and submersed in refolding buffer (20 mM Tris, 5 mM CaCl_2_, 1 µM ZnCl_2_, pH 8.0) for 20 h at 37 °C. Gels were dyed using Coomassie Brilliant Blue overnight and de-stained for band revelation [77]. Gels were documented (ChemiDoc, Bio-Rad), and clear bands were quantified on FIJI [67]. For quality control, a loading sample was used in all gels to reduce differences in dye efficiency. Each sample was run twice in different gels (*n* = 3).

### 4.13. Integrin Immunophenotyping by Flow Cytometry

Cells were scraped in ice-cold PBS, pelleted by centrifugation (8 min at 1200 rpm, 4 °C), and resuspended in PBS for binding with primary antibodies: anti-integrin β1 (1 µg/sample, 13,590 Santa Cruz Biotechnology) and anti-integrin β3, (0.125 µg/sample, 110,131 Abcam). After 1 h incubation at 4 °C, tubes were washed twice with PBS (1200 rpm, 10 min, 4 °C) before secondary antibody probing (45 min, 4 °C—Goat Anti-Mouse IgG H&L [FITC], 1:1000, 6785 Abcam). Samples were washed twice with PBS (1200 rpm, 10 min, 4 °C) and resuspended for flow cytometry analysis on BD Accuri^TM^ C6 (BD Biosciences). A total of 15,000 events were captured in each replicate (*n* = 3). Doublets were removed using the standard gating technique (Appendix A), and singlets were displayed in histograms for median determination. Population comparison was performed on FlowJo (v.10.8.1) using the super enhanced Dmax subtraction (SE Dymax) as the comparative parameter.

### 4.14. Matrigel Invasion Assay

An invasion chamber was assembled using 8.0 µm-pore inserts (662,638 Greiner) coated with Matrigel (Corning) for 2 h at 37 °C, installed in a 24-well plate (662160-01 Greiner). Cells (10^5^/well) were treated with SEVh (5 µg/mL) in FBS-free media on ice and applied to the top chamber. Inserts were assembled atop wells containing DMEM 10% FBS (negative control: FBS-free DMEM) and incubated for 6 and 16 h in normoxia or hypoxia (1% O_2_). Membranes were fixed with 4% PFA for 10 min, washed, and cleaned with a cotton swab. Cell nuclei were stained with DAPI (0.7 ng/µL) for 10 min. Membranes were detached and assembled in slides for epifluorescence imaging (ImageXpress Micro, Molecular Devices) at 10× magnification, followed by an automated nuclei count. The cell invasion index was calculated as the ratio between the total nuclei count from groups and the total nuclei count from untreated, normoxic cells.

### 4.15. Fluorescent Gelatin Degradation Assay

Round coverslips were coated with poly-L-lysine (0.1 mg/mL) for 20 min, RT, cross-linked with 0.5% glutaraldehyde for 15 min, and covered with a thin layer of fluorescent gelatin (G13187 Molecular Probes, 0.2 mg/mL) overnight at 4 °C. MDA-MB-231 (50,000 cells/well) were treated with SEVh (5 µg/mL) and plated in FBS-free DMEM (positive control: DMEM 10% FBS). Cells were incubated for 24 and 48 h in normoxia or hypoxia, protected from light. Cells were fixed with 4% PFA for 10 min, permeabilized for 5 min with 0.1% Triton X-100, and blocked for 1 h with 1% BSA-PBS, RT. Tubulin was immunoprobed (5 µg/mL; 80,779 Abcam) overnight at 4 °C, followed by a 1-h incubation with the secondary antibody AlexaFluor 647 (1 µg/mL; 150,115 Abcam). DAPI (0.7 ng/µL) was used for nuclei staining. Slides were assembled using Prolong^TM^ mounting media (Thermo Fisher Scientific). Epifluorescence images were acquired for field analysis (ImageXpress Micro, Molecular Devices) under 20× magnification, and the representative imaging was performed under 60× magnification in a confocal microscope (Olympus FV10i). We quantified the total degraded area over a 72 × 72 µm frame and then normalized the values by cell count.

### 4.16. Cell Morphology Assay

Cells were plated on round uncoated plates or coverslips coated with gelatin (0.2 mg/mL) or Matrigel (1:1, *v*/*v*). After incubation in normoxia or hypoxia for 24 h or 48 h, samples were fixated with 4% PFA for 10 min, RT, washed, and permeabilized with 0.1% Triton X-100 for 5 min, RT. Cell nuclei and F-actin were stained with DAPI (0.7 ng/µL) + 2.5% Phalloidin-FITC for 20 min. Image acquisition was performed at 20× magnification (IN Cell Analyzer 2200, GE Healthcare, Chicago, IL, USA). The parameters of cell circularity, aspect ratio (AR), and perimeter/area ratio (Appendix A) were calculated on FIJI [67] (*n* = 300).

### 4.17. Apoptosis Assay

Cells were centrifuged (400× *g*, 5 min, 4 °C), washed on iced PBS (2000× *g,* 5 min, 4 °C), and scrapped in binding buffer for staining with Annexin V-PE and 7AAD (7-aminoactinomycin) using the BD Pharmingen^TM^ PE Annexin V Apoptosis Detection Kit I (Cat 559,763, Lot 8,086,787, BD Biosciences). Cells were centrifuged (2000 rpm, 5 min, 4 °C) and resuspended in binding buffer for flow cytometry analysis on BD Accuri^TM^ C6 (BD Biosciences), where 15,000 events were analyzed. Experimental controls included unstained and untreated cells in both incubation settings and single-stained and double-stained controls (Appendix A). The assay was conducted with technical duplicates and biological triplicates.

### 4.18. Total RNA Extraction and cDNA Synthesis

Cells were scraped in ice-cold TRIzol (Invitrogen) and mixed with chloroform at RT for 15 min. After centrifugation (12,000× *g*, 10 min, 4 °C), the upper translucent fraction was collected and mixed with isopropanol for 10 min, RT. Samples were centrifuged at 12,000 rpm for 10 min at 4 °C and washed twice with 75% ethanol (7500 rpm, 5 min, 4 °C). After dried, pellets were resuspended in ultrapure water and quantified (NanoDrop 2000, Thermo-Fisher Scientific). RNA samples were treated with DNase I (#18068-015, Invitrogen), and reverse transcription was performed using the High-capacity cDNA Reverse Transcription kit (Applied Biosystems, Waltham, MA, USA).

### 4.19. Gene Expression by RT-qPCR

Primers for the analysis of HIF-1α (NM_001243084.2), MMP-2 (NM_004530.4), MMP-9 (NM_004994.2), integrin β1 (NM_133376.3), and integrin β3 (NM_000212.3) expression by qPCR were designed using the online tool (http://www.ncbi.nlm.nih.gov/tools/primer-blast, accessed on 8 February 2021). DNA sequences were checked using Blast (http://blast.ncbi.nlm.nih.gov/Blast.cgi, accessed on 8 February 2021). The primer sequences for targeted genes were: HIF1A (F: AAAATCTCATCCAAGAAGCC; R:AATGTTCCAATTCCTACTGC), ITGB1 (F: ATTCCCTTTCCTCAGAAGTC; R: TTTTCTTCCATTTTCCCCTG), ITGB3 (F: CTCCGGCCAGATGATTC; R: TCCTTCATGGAGTAAGACAG), MMP2 (F: AGGACCGGTTCATTTGGCGG; R: TGGCTTGGGGTACCCTCGCT), and MMP9 (F: CGCTACCACCTCGAACTTTG; R: GCCATTCACGTCGTCCTTAT). Housekeeping genes GAPDH (F: ACAGTTGCCATGTAGACC; R: TTGAGCACAGGGTACTTTA) and HPRT (F: TGACACTGGCAAAACAATGCA; R: GGTCCTTTTCACCAGCAAGCT) were used as normalizers. Reactions were set to start at 95 °C for 10 min, followed by 40 cycles of 95 °C for 15 s; 59–68 °C for 30 s; and 72 °C for 30 s. The melt curve was produced by increasing the temperature from 65 °C to 95 °C in 0.5 °C increments every 30 s. PCR products were determined based on Ct values (threshold cycle), in which each gene expression variation was equal to 2^−∆∆Ct^ [78].

### 4.20. Statistical Analysis

All assays were performed in technical triplicates and at least three independent experiments. Data were submitted to ROUT’s outlier detection test and normality tests (Shapiro-Wilk for *n* < 9, D’Agostino-Pearson omnibus K2 for *n* ≥ 9). In case of lognormality, datasets were transformed into log_10_. Parametric data were analyzed using ANOVA one-way with Tukey’s multiple comparison post-hoc. Non-parametric data were tested using the Kruskal-Wallis analysis of variance, with Dunn’s multiple comparison post-test. Values of *p* < 0.05 were considered statistically relevant. Results are presented as mean ± standard deviation (SD), standard error of mean (SEM) if parametric, or median ± interquartile range if non-parametric. Data analysis and graph design were made on GraphPad Prism (v. 9.3).

## 5. Conclusions

SEV separated from the TNBC cell line MDA-MB-231 in hypoxia promotes invasive behaviors in normoxia, while inducing proteolytic and catabolic pathways in hypoxia. Cell invasion is favored by SEVh under advantageous conditions of oxygen and matrix composition due to the modulation of gelatinase (i.e., MMP-2 and MMP-9) and integrin (i.e., subunits β1 and β3) expression and availability. These modifications interfere with the F-actin cytoskeleton to create a migratory phenotype coupled with invadopodia development. Protein profiling of SEVh-treated, 20% O_2_-cultured cells indicate a baseline favoring metabolic processes and cell cycles, modulating cell health away from apoptotic pathways that are enriched in hypoxia. Overall, our results demonstrate the importance of hypoxic signaling via SEV in a tumoral setting for the establishment of metastasis.

## Figures and Tables

**Figure 1 ijms-23-12646-f001:**
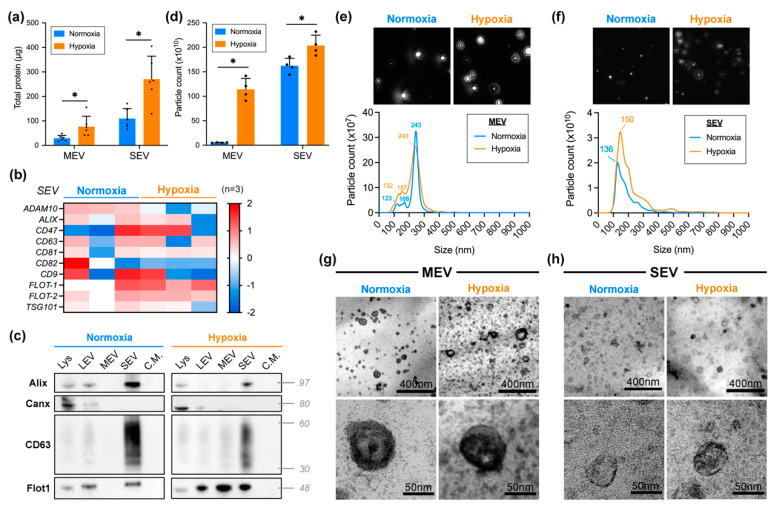
Isolation and characterization of EVs. (**a**) Total protein content for normoxic (**blue**) and hypoxic (**orange**) MEV and SEV. Data presented as mean ± SD (standard deviation). (**b**) SEV biomarker profile obtained via proteomic analysis. Each column indicates a biological replicate (*n* = 3). (**c**) Immunoblotting membranes probed for proteins Alix, calnexin (Canx), CD63, and flotillin-1 (Flot1) containing samples of cell lysate (Lys), LEV, MEV, SEV, and the leftover conditioned media (C.M.) of EV isolations. (**d**) Particle count for MEV and SEV (mean ± SD). (**e**,**f**) Mean vesicle size dispersion for (**e**) MEV and (**f**) SEV (*n* = 5). (**g**,**h**) Transmission electron microscopy images of broad field and individual (**g**) MEV and (**h**) SEV. Scale bar: 400 nm (**top**), 50 nm (**bottom**) * *p* < 0.05.

**Figure 2 ijms-23-12646-f002:**
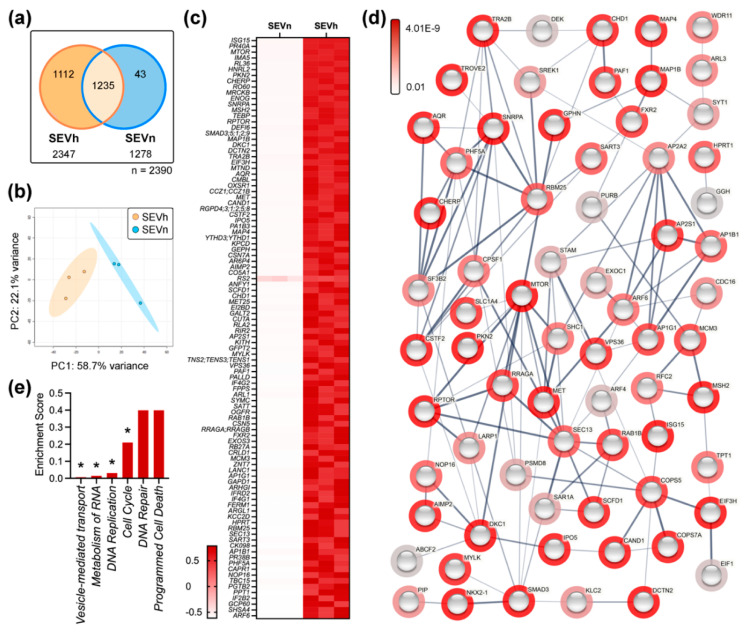
Comparative SEV proteomic profiling. (**a**) Venn diagram of protein distribution in SEV. (**b**) Principal component analysis (PCA) score plot. (**c**) Heatmap of top 100 proteins differentially expressed in SEVh (*p* < 0.01). (**d**) Interaction map of top 144 differentially expressed proteins in SEV samples with hidden disconnected nodes, created on String-DB. Halos indicate the strength of *p* values from the statistical analysis of normalized data (red to gray). Strings are shown as confidence network edges, with soft lines for medium confidence (0.4), regular lines for high confidence (0.7) and bold lines for highest confidence (0.9). (**e**) Enrichment analysis on Reactome indicates the upregulation of pathways from vesicle-mediated transport, metabolism of RNA, DNA replication, and cell cycle in SEVh compared to SEVn. Lower scores are indicative of higher statistical significance. * *p* < 0.05.

**Figure 3 ijms-23-12646-f003:**
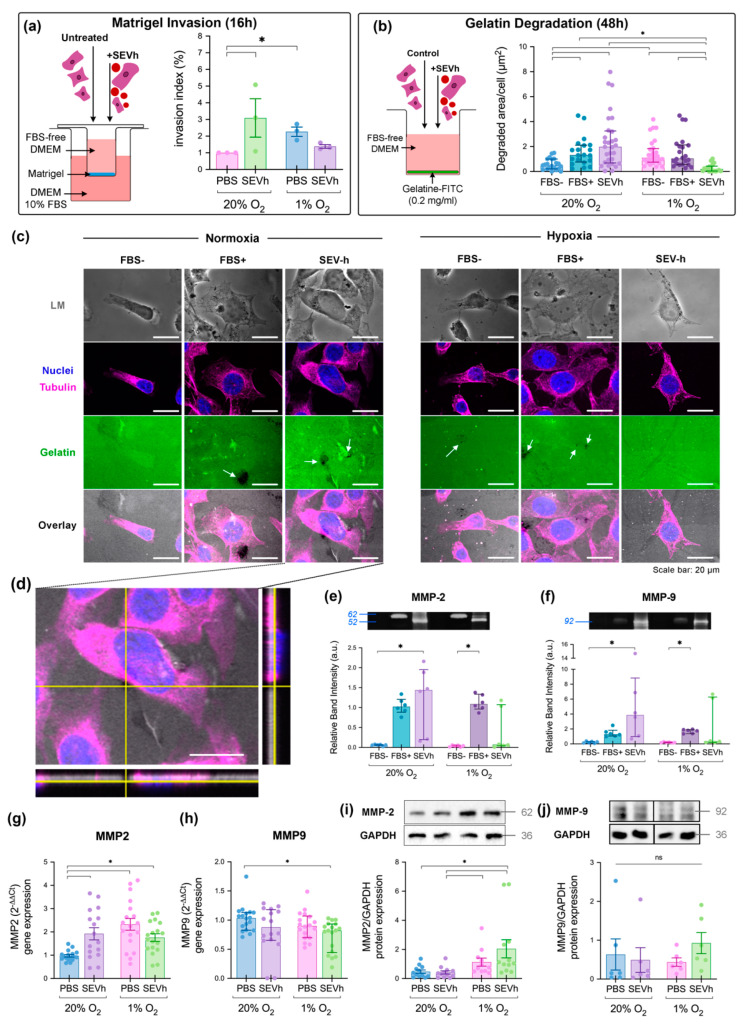
Normoxia triggers SEVh-mediated long-term invasion. (**a**) Experimental diagram of the Matrigel invasion chamber assay (left) and invasion index (%) after 16 h (median ± interquartile, right). (**b**) Experimental diagram of the fluorescent gelatin degradation assay (left) and comparison of the degraded area after 48 h (median ± interquartile, right). (**c**) Representative images of the gelatin degradation assay in light microscopy (LM) and fluorescent nuclei (DAPI, blue), tubulin, (AlexaFluor647, pink) and gelatin (0.2 mg/mL, fluorescein, green). Below, there are the merged images. Degradation spots in black are indicated by white arrows (scale bar: 20 µm). (**d**) Orthogonal view of normoxic, SEVh-treated cells with the degradation spot near the cellular body (scale bar: 20 µm). (**e**,**f**) Proteolytic activity from the gelatin degradation assay supernatant of (**e**) MMP-2 (mean ± SEM) and (**f**) MMP-9 (median ± interquartile). (**g**,**h**) Gene expression of (**g**) MMP2 (mean ± SEM) and (**h**) MMP9 (median ± interquartile) in MDA-MB-231 cells. (**i**,**j**) Protein expression of (**i**) MMP-2 (mean ± SD) and (**j**) MMP-9 (median ± interquartile). * *p* < 0.05.

**Figure 4 ijms-23-12646-f004:**
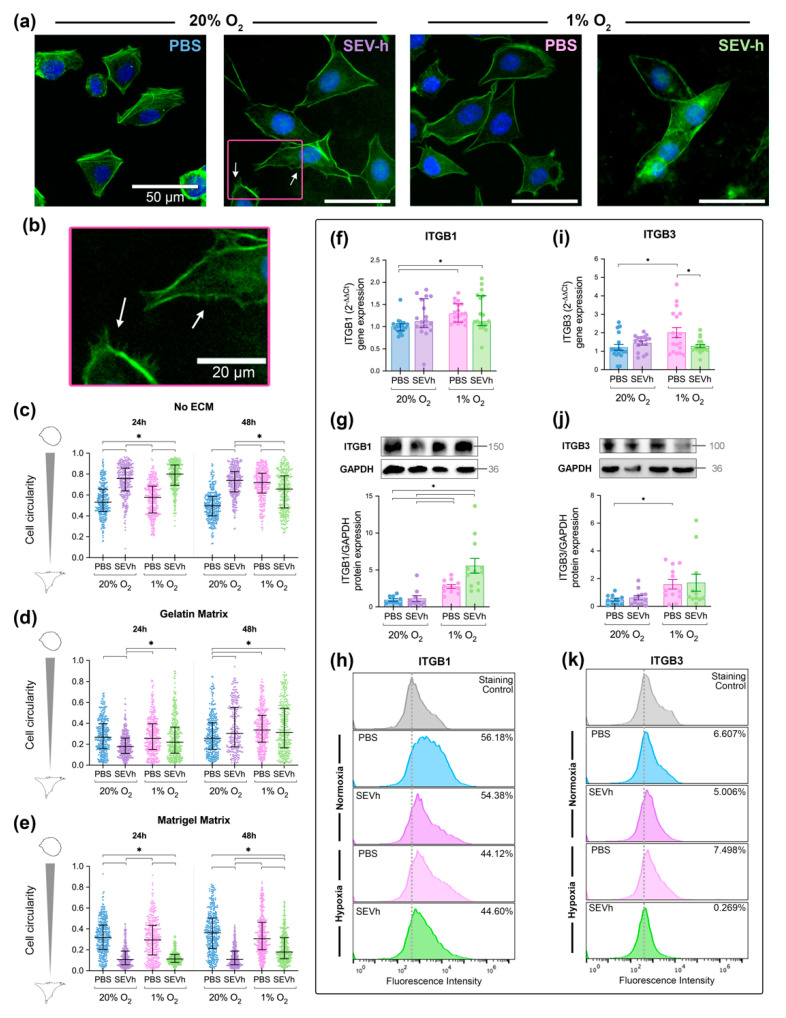
Phenotypical alterations promoted by hypoxia and/or SEVh exposure. (**a**) Representative images of untreated (PBS) and treated (SEVh) MDA-MB-231 cells in Matrigel matrix following a 24 h incubation in normoxia (20% O_2_) or hypoxia (1% O_2_). White arrows indicate the presence of invadopodia, indicating a migratory phenotype (scalebar: 50 µm). (**b**) Detailed vision of invadopodia in normoxic cells following SEVh exposure (scalebar: 20 µm). (**c**–**e**) Circularity determination after 24 h and 48 h of cells plated atop (**c**) an uncoated surface, (**d**) gelatin (0.2 mg/mL) matrix, and (**e**) Matrigel matrix (1:1, *v*/*v*). (**f**) Gene expression (median ± interquartile), (**g**) whole cell protein expression (mean ± SEM), and (**h**) surface protein expression of integrin subunit β1 (data presented as mean % of triplicates of two independent assays). (**i**) Gene expression (mean ± SEM), (**j**) whole cell protein expression (mean ± SEM), and (**k**) surface protein expression of integrin subunit β3 (data presented as mean % of triplicates of two independent assays). * *p* < 0.05.

**Figure 5 ijms-23-12646-f005:**
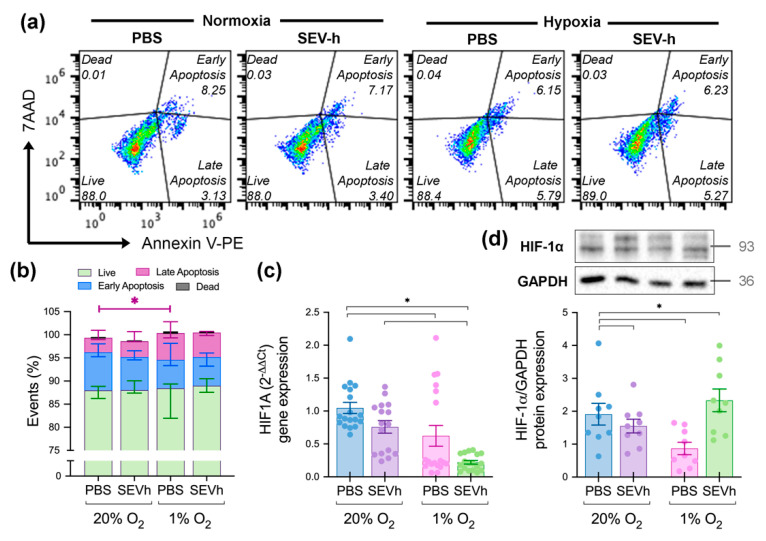
HIF-dependent cell responses resulted from SEVh treatment. (**a**) Scatter plot of apoptosis assay; population (%) displayed as average of two independent assays in technical triplicates. (**b**) Comparative graph of the four possible outcomes for cells (live, early apoptotic, late apoptotic, or dead) as determined by the apoptosis assay. Statistical significance obtained from early apoptotic cells between untreated groups (normoxia × hypoxia), as indicated in the fuchsia bar above (**c**,**d**) Levels of HIF-1α (**c**) gene expression (mean ± SEM) and (**d**) whole cell protein expression (mean ± SEM) (* *p* < 0.05).

**Figure 6 ijms-23-12646-f006:**
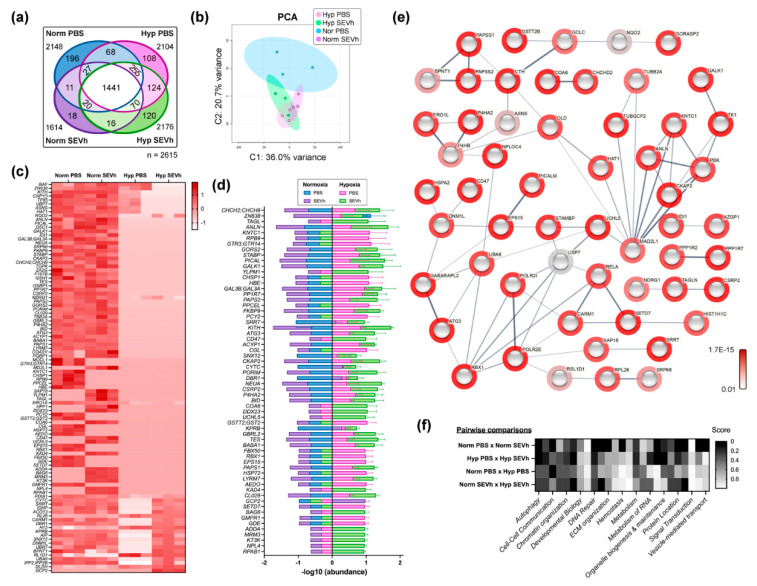
Comparative proteome analysis. (**a**) Venn diagram of protein distribution in MDA-MB-231, noting cells incubated in normoxia (Norm) or hypoxia (Hyp) in the presence or absence of SEVh. (**b**) Principal component analysis (PCA) score plot. (**c**) Heatmap of 105 proteins differentially expressed in cells treated or not with SEVh, in normoxia and hypoxia (*p* < 0.05). (**d**) Values of −log_10_ (abundance) of the 62 most divergent proteins differently expressed between groups (*p* < 0.01). (**e**) Interaction map of all differentially expressed proteins in cellular samples with hidden disconnected nodes created at String-DB. Halos indicate the strength of *p* values from the statistical analysis of normalized data (red to gray-scale below). Strings shown as confidence network edges, with soft lines for medium confidence (0.4), regular lines for high confidence (0.7), and bold lines for highest confidence (0.9). (**f**) Heatmap containing the enrichment analysis of statistically distinct proteins by pairwise comparisons, developed in Reactome. Upregulated pathways are indicated below, with significance varying from the most (black) to least (white) enriched in each comparison, following the scale (left—*p* < 0.05).

## Data Availability

Detailed experimental data for EV isolation and characterization are available on the EV-TRACK platform [25] (EV-TRACK ID: EV220177). Data are available via ProteomeXchange with identifier PXD035244. Raw data from other assays are available on demand.

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
