# Peer review of "Small Extracellular Vesicles from Hypoxic Triple-Negative Breast Cancer Cells Induce Oxygen-Dependent Cell Invasion"

_ijms, 2022, doi:10.3390/ijms232012646_

Round 1
Reviewer 1 Report
The authors report small extracellular vesicles inducing invasive properties of triple-negative breast cancer cells.
In general, the topic is very interesting and of high significance. The paper is well structured.
However, there is some minor issues that the authors may want to consider before publication.
1. Why do the authors write about MEV only in Section 3.1? According to the results of the nanoparticle tracking analysis, SEV preparations may also contain MEV. Consequently, these vesicles MEV may also induce invasive cell properties. How was it revealed that only SEV induced the invasive properties of cells? Have the authors investigated the role of MEV in the invasive properties of cells for control?
As recently published by Jeppesen et al (Cell 2019, PMID:30951670), highly purified SEV preparation requires the sequential combinations of different methods (ultracentrifugation, high-resolution density gradient fractionation, and direct immunoaffinity capture). Thus, it should be mentioned that the purity of the SEV preparation in this study is nonoptimal.
2. The scale bar should be on all photos of transmission electron microscopy (Fig. 1 (g) and (h)), fluorescence microscopy (Fig. 3 (c), (d). Fig. 4 (a)).
3. Please add punctuations (Fig.2).
Author Response
We thank the reviewer for the pertinent observations and critics, and we are providing a point-by-point response below.
1. We wrote about MEV only in Section 3.1 as a means to fully characterize our samples, indicating that our EV subpopulations (MEV and SEV) from both oxygen conditions have distinct characteristics among themselves. Even though there are records in the literature indicating the invasive properties of MEV (Das 2019, PMID: 31341019), we decided to focus only on the study of hypoxic SEV as a means to understand the role of this EV subtype during tumor cell invasion.
We are aware that no single method for EV separation guarantees the obtention of a pure EV subpopulation, and that additional methods may be used to increase purity, however even when applying a secondary separation method EV samples may be contaminated with protein aggregates, lipid rafts and other EV subtypes or non-EV components (Willms 2018, PMID: 29760691). Hence, we did not assume to work with a purified SEV sample, but with a sample that was enriched with SEV and that differed from its previous isolate (MEV). Nevertheless, we agree that this topic should be brought up in our discussion.
2. All microscopy images were replaced to contain scale bars.
3. We apologize for the mistake and have punctuated the figure subtitles.
Reviewer 2 Report
The manuscript by Pachane and colleagues examines the effect of extracellular vesicles (EV) from breast cancer cells under hypoxia on breast cancer cell’s invasiveness. EV from hypoxic and normoxic MDA-MB-231 breast cancer cells were collected, then collected EV were used to treat MDA-MB-231 cells and analyzed for cell physiological changes. EV’s were analyzed for their protein content, with significant proteomic differences being found between normoxic and hypoxic derived EVs. Hypoxia derived EVs increased Matrigel invasion and gelatin degradation, but only under normoxic conditions, while causing the opposite effect under hypoxic conditions. EVh also did not change expression of integrins or change hypoxia-induced apoptosis but increased HIF-1 protein expression under hypoxic conditions. Lastly the authors present data of their proteomic analysis comparing MDA-MB-231 cells under normoxic or hypoxic conditions, with or without EVh treatment. The largest difference in protein expression were observed between normoxic and hypoxic cells, which smaller changes with EVh treatment under hypoxic conditions.
General comments:
The authors curated a lot of data from a relatively small experiment. It is unfortunate that the results are based on a single cell line, which vastly decreases the confidence in the finding’s relevance to the overall behavior of breast cancer cells.
The premise of the study is not well introduced. It seems strange that EV derived from one cell type would act on the same cell type. The literature that introduces the topic seems to indicate that EV primarily act long-distance and on different tissues. I would be curious as to the authors’ reasoning to examine EV acting on their secreting cell line similar to autocrine hormonal mechanisms. Likewise, the sequence of treatments is hard to imagine following physiological processes, where cells are under hypoxic conditions initially, excreting EVs, then under normoxic conditions as they are taking in the EVs they initially produced. It would be great to have some more clarification on why the authors decided to model this sequence of events in their experiments.
It is very hard to follow the authors intentions throughout their discussion. For example I would agree with the statement that the integrin modifications they observed are due to the hypoxic environment rather than signaling from EVh, however the next statement indicates that EVhs modulate integrins availability in the cell surface, which does not seem to be supported by any of their data.
Specific comments:
Could the authors clarify their EV treatment sequence in 2.6. Cells were treated 24h with or without hypoxia initially and then an additional 24h with EV. Was the additional 24h under the same hypoxic conditions or were all cells kept at normoxic conditions for the additional 24h.
Figure 1b displays dramatic differences between the repeats. CD47 looks to have the highest upregulation and downregulation within the normoxic EVs. CD82 seems to have three different responses within the normoxic group. Do the authors have some data on CI values within groups?
In Figure 3j the representative Blot shown is not representative of the quanitified data in the corresponding graph. MMP9 expression in the normoxic sEVh treated samples seems to be considerable higher than the PBS control as both have comparable GAPDH expression.
The results in Figure 5c/d are intriguing but may need better explanation. It seems counterintuitive that hypoxia alone should decrease HIF-1 expression in these cells. Here is where a second cell lines may have given another indication what is going on. In the discussion the authors mention that in their data they find an enrichment in ubiquitination, but HIF-1 ubiquitination levels are not examined in this study. Also, since hypoxia should regulate HIF-1 levels on the protein levels, the finding that HIF-1 levels change on the gene expression level should be discussed.
Author Response
We thank the reviewer for the relevant observations and critics, and we are providing a point-by-point response below.
General comments:
1. We agree that the results based on a single cell line are rarely indicative of an overall behavior of breast cancer cells, especially considering its intra- and inter-tumoral heterogeneity (Aw Yong 2020, PMID: 32238832). Yet we decided to publish our findings considering their relevance to understanding autocrine-like signaling within a tumor setting, where hypoxia is established.
2. We have updated our introduction to include the premise of the study, in hopes of enlightening the readers to our reasoning. Our experimental design is distinct from most works in the literature because we wanted to understand how intratumoral communication might occur within a single tumoral cell type considering hypoxia as the only variable. Also, although EVs have a crucial role in long-distance communication, it is expected that some EV will be uptaken by the surrounding cells as they are secreted. This short-distance, autocrine-like signaling is seldom discussed in the literature, hence our interest in investigating which cell behaviors can be triggered by EVh and/or hypoxia.
3. We have also re-evaluated our discussion to remove inconsistencies, and we apologize for our mistakes.
Specific comments:
4. In our experimental design, cells were incubated in normoxia or hypoxia for 24h immediately after plating to allow cell adhesion in different oxygen conditions. After this, cells were treated with previously isolated SEVh and incubated again in the same conditions, normoxia or hypoxia, for further 24h. We have re-written this section for better understanding.
5. We agree that there are differences between repeats, as expected from biological samples, yet none of the biomarkers were statistically different between normoxic and hypoxic EVs. Calculating the 95% CI of normalized samples [log10(abundance)], we determined that the confidence limit of samples were: 0.4477 to 0.4814 (mean #1: 0.4645), 0.2210 to 0.2550 (mean #2: 0.2380) and 0.3471 to 0.3996 (mean #3: 0.3734) for SEVh repetitions, and -0.2047 to -0.1649 (mean #1: -0.1848), -0.7138 to -0.6691 (mean #2: -0.6915) and -0.2179 to -0.1814) (mean #3: -0.1997) for SEVn replicates. We believe that differences are acceptable, according to this analysis.
6. We apologize for this mistake. We have reanalysed the data and replaced the blot to a better representative image.
7. We agree that this result needs to be discussed further, and we have added it to our manuscript.
Round 2
Reviewer 1 Report
The authors provided a significantly improved version.
Author Response
Thank you for your considerations.
Reviewer 2 Report
The authors have addressed my concerns satisfactorily.
Author Response
Thank you for your considerations.